# Optimized Machine Learning Models for Predicting Core Body Temperature in Dairy Cows: Enhancing Accuracy and Interpretability for Practical Livestock Management

**DOI:** 10.3390/ani14182724

**Published:** 2024-09-20

**Authors:** Dapeng Li, Geqi Yan, Fuwei Li, Hai Lin, Hongchao Jiao, Haixia Han, Wei Liu

**Affiliations:** 1Poultry Institute, Shandong Academy of Agricultural Sciences, Jinan 250100, China; lidp1990@126.com (D.L.); lifuwei1224@163.com (F.L.); hanhaixia@163.com (H.H.); 2Shandong Provincial Key Laboratory of Livestock and Poultry Breeding, Jinan 250100, China; 3College of Animal Science and Technology, Shandong Agricultural University, Taian 271018, China; yangeqi@sdau.edu.cn (G.Y.); hailin@sdau.edu.cn (H.L.); hongchao@sdau.edu.cn (H.J.)

**Keywords:** precision livestock management, animal welfare, thermal comfort, core body temperature, machine learning, optimization algorithm, SHAP value

## Abstract

**Simple Summary:**

In hot weather conditions, ensuring dairy cow comfort and preventing heat stress is crucial. This study applied machine learning techniques to predicting dairy cows’ core body temperatures, and improved prediction accuracy through data preprocessing, feature engineering, and hyperparameter optimization. This facilitates timely actions, such as enhancing ventilation or implementing mist cooling, to maintain the health and productivity of the cows. By enhancing the accuracy and interpretability of predictions, the study provides a powerful tool for precision livestock management, contributing to improved animal welfare and enhanced economic farm efficiency.

**Abstract:**

Heat stress poses a significant challenge to livestock farming, particularly affecting the health and productivity of high-yield dairy cows. This study develops a machine learning framework aimed at predicting the core body temperature (CBT) of dairy cows to enable more effective heat stress management and enhance animal welfare. The dataset includes 3005 records of physiological data from real-world production environments, encompassing environmental parameters, individual animal characteristics, and infrared temperature measurements. Employed machine learning algorithms include elastic net (EN), artificial neural networks (ANN), random forests (RF), extreme gradient boosting (XGBoost), light gradient boosting machine (LightGBM), and CatBoost, alongside several optimization algorithms such as Bayesian optimization (BO) and grey wolf optimizer (GWO) to refine model performance through hyperparameter tuning. Comparative analysis of various feature sets reveals that the feature set incorporating the average infrared temperature of the trunk (IRTave_TK) excels in CBT prediction, achieving a coefficient of determination (R^2^) value of 0.516, mean absolute error (MAE) of 0.239 °C, and root mean square error (RMSE) of 0.302 °C. Further analysis shows that the GWO–XGBoost model surpasses others in predictive accuracy with an R^2^ value of 0.540, RMSE as low as 0.294 °C, and MAE of just 0.232 °C, and leads in computational efficiency with an optimization time of merely 2.41 s—approximately 4500 times faster than the highest accuracy model. Through SHAP (SHapley Additive exPlanations) analysis, IRTave_TK, time zone (TZ), days in lactation (DOL), and body posture (BP) are identified as the four most critical factors in predicting CBT, and the interaction effects of IRTave_TK with other features such as body posture and time periods are unveiled. This study provides technological support for livestock management, facilitating the development and optimization of predictive models to implement timely and effective interventions, thereby maintaining the health and productivity of dairy cows.

## 1. Introduction

The core body temperature (CBT) of dairy cows, as one of the most fundamental physiological indicators, is frequently employed to evaluate the level of thermal stress they experience due to environmental factors [1]. Under normal circumstances, the CBT of dairy cows is maintained within the range of 38.3 to 38.7 °C. In production settings, timely monitoring of this vital sign and implementing effective cooling measures, such as misting and enhanced ventilation, before or upon deviation from the norm, is crucial for safeguarding the health and ensuring the productive capacity of dairy cows [2].

Presently, the primary method for measuring CBT in dairy cows involves rectal or vaginal temperature readings [3], which are not only time-consuming and labor-intensive but also incapable of continuous monitoring. While continuous monitoring systems have been developed, they are predominantly invasive, potentially causing stress to the animals, and are limited in their applicability in large-scale farming environments [4]. As such, developing a non-invasive, real-time, and practical predictive model presents itself as an attractive solution. Numerous studies have confirmed that thermal environmental factors directly impact the CBT of dairy cows, as evidenced by Kim et al.’s observation in climate-controlled experiments that rectal temperatures of Holstein cows increased by 0.4 and 0.9 °C when daytime temperatures rose from 25 to 30 °C [5]. Zhou et al. [6] further demonstrated that the inflection point temperature for rectal temperature increases in dairy cows decreases with higher relative humidity levels while increasing with faster air velocities. Additionally, the production traits and behavioral activities of dairy cows are also found to be closely associated with fluctuations in their CBTs [7,8,9]. Currently, predictive models for dairy cow CBT are predominantly built using multivariate regression analysis of environmental factors. For instance, Li et al. [10] constructed a model for predicting the respiration rate of dairy cows using environmental temperature, relative humidity, wind speed, milk yield, and time interval as input parameters, albeit limited to predicting mean CBTs during specific daytime periods. There are also studies that employ thermoregulatory models based on heat balance principles for hourly predictions of an individual cow’s core body temperature, though the predictions are heavily dependent on the initial settings of environmental and physiological parameters [11,12].

In recent years, propelled by the rapid advancements in artificial intelligence technologies, data-driven machine learning methods have been successfully applied to the prediction of dairy cow diseases and production performance [13]. Compared to traditional multivariate regression, machine learning models excel at capturing complex nonlinear relationships among variables. From 2014 to 2023, some progress was made in applying machine learning techniques to predict dairy cow core body temperatures, encompassing studies from various regions around the globe [14,15,16,17], which are summarized in Table 1. These predictive models typically integrate environmental factors (such as temperature, humidity, and wind speed), animal behavioral indicators (such as respiration rate and posture), and production data (such as milk yield). On the algorithmic front, explorations have ranged from linear regression to artificial neural networks, and even to ensemble learning techniques such as XGBoost and CatBoost. Among them, artificial neural networks have particularly stood out in their performance in predicting dairy cow core body temperatures. Nevertheless, there remain limitations in the application of machine learning methods in this field, primarily due to constraints in data availability, leading to underexplored key predictive factors. For example, infrared thermography, a non-invasive temperature measurement technique, has been widely used to investigate the correlation between surface and core temperatures in animals [18,19,20]. Moreover, the close relationship between the thermal comfort indices—a composite result of multiple environmental factors—and dairy cow thermal physiological indicators has been validated. The dynamic changes in body temperature reflect the thermal exchange processes between animals and their environment [21], suggesting that certain heat exchange quantities may serve as potential predictive indicators. Hyperparameters in machine learning models, which are manually set before training to control the learning process and complexity, significantly influence the model’s performance and generalization capabilities. In previous research, hyperparameter optimization was often overlooked, and even when utilized, the optimization strategies might have been limited, such as random search or grid search being inefficient in high-dimensional parameter spaces. Deep hyperparameter tuning requires substantial computational resources and time, restricting the breadth and depth of combinations that researchers can explore. Furthermore, the issue of model interpretability becomes increasingly pronounced, especially with “black box” models such as artificial neural networks, whose internal workings are opaque, complicating the understanding of the decision-making process.

In light of this, our study aims to utilize machine learning algorithms to construct a predictive model for dairy cow core body temperature, extracting readily available data from real-world production environments, and identifying and integrating features that can significantly enhance predictive accuracy. We will also apply optimization algorithms to fine-tune the model’s hyperparameters, compare and analyze the performance versus cost-effectiveness of optimization, and ultimately employ models with strong interpretability to enhance transparency and reliability. This endeavor is expected to provide scientific support for the management of dairy cow thermal stress, thereby promoting more precise and efficient livestock management practices.

## 2. Materials and Methods

As illustrated in Figure 1, this study presents a generalized research framework for developing CBT prediction models for dairy cows and carrying out SHAP-based feature analysis. This generalized framework consists of three parts: (1) the establishment of a database, involving data collection, feature creation, and data processing; (2) the development of predictive models, including the selection of machine learning algorithms, hyperparameter optimization algorithms, model training and evaluation, and the choice of the optimal model; (3) feature analysis based on SHAP, which identifies the key drivers affecting CBT prediction through the SHAP summary plot, clarifies how the impact of one feature on CBT interacts with another feature via the SHAP dependence plot, and offers local explanations by using the SHAP waterfall plot. This research framework is not confined to the study of the CBT of dairy cows but is universal and can be applied to the prediction of other physiological indicators associated with the thermal state of dairy cows.

### 2.1. Database Establishment

#### 2.1.1. Data Collection

The dataset utilized in this study originated from a commercial dairy farm in northern China (coordinates: 39°13′ N, 117°2′ E), characterized by a temperate continental monsoon climate. The data were recorded from July to October 2020. Throughout the trial period, there were a total of 30 experimental days, with an interval ranging from 2 to 4 days between each pair of experimental days. On each experimental day, 90 to 120 Holstein dairy cows were randomly observed, and a cumulative total of 826 cows were monitored throughout the experimental days. Each cow was inspected only once on each experimental day and was observed between 1 and 10 times throughout the entire trial period. The cows were housed in adjacent and identically conditioned dairy barns. Each barn measured 107 m in length and 31 m in width. It adopted a four-row head-to-head arrangement, and the roof was of a double-slope type. The barn was equipped with fan-sprinkler cooling. When the ambient temperature exceeded 18 °C, the turbulent fans (with an air volume of 25,430 m^3^/h and a longitudinal arrangement spacing of approximately 6.0 m) were activated. When the temperature–humidity index (THI) surpassed 72, the spray cooling was initiated. During the experiment, milking, feeding, and manure cleaning of the cows were conducted in accordance with the normal production management of the ranch.

The collected environmental parameters encompassed air temperature (Ta, °C), relative humidity (RH, %; HOBO U23-001, Onset Computer Corporation, Bourne, MA, USA, with an accuracy of ±0.2 °C and ±2.5%), black globe temperature (Tbg, °C; JTR04, JantyTech Inc., Fengtai, Beijing, China, with an accuracy ±0.6 °C), wind speed (U, m/s; TSI 9565P, TSI Corporation, Shoreview, MN, USA, with an accuracy of ±0.015 m/s), and solar radiation intensity (Qsr, W/m^2^; TES 1333R, TES Electronic Corporation, Tapei, China, with an accuracy of ±10 W/m^2^); the sampling interval was 10 min. To prevent interference with milking (at 07:30, 13:30, and 19:30 daily), the CBTs were collected within the time frame of 8:00–12:00 and 14:00–18:00 of each experimental day by inserting a veterinary digital thermometer (manufactured by Shangnong Technology Co., Ltd., Qingdao, China; with a temperature accuracy of ±0.2 °C and a measurement range of 32.0–42.0 °C) into the rectum of the cows at approximately 10 cm. After collecting the rectal temperatures, the surface temperatures on the side of the cows away from the fan were measured using an infrared imager (Fotric 235, Fortic Inc., Jing’an, Shanghai, China). This imager boasts a 336 × 252 image resolution, a thermal sensitivity of <50 mk NETD (Noise Equivalent Temperature Difference) @30 °C, a spatial resolution quality < 1.27 mrad IFOV (Instantaneous Field of View), and an accuracy of ±2 °C or ±2%, which meets the minimum specifications recommended in the Veterinary Thermal Imaging Guidelines [22]. The distance between the cows and the observers was 1.5 m. The individual production data such as the parity, milk yield, and lactation days of the cows were retrieved through the management software (Afifarm, Afimilk Ltd., Kibbutz, Israel).

#### 2.1.2. Feature Creation 

The predictive potential of a machine learning-based model can be enhanced through the selection and creation of some valuable features, which are illustrated in Figure 2.

Thermal comfort indices

Thermal comfort indices, derived from environmental data, widely serve as a proxy for the cow’s thermal stress level [23]. Previous studies have evaluated the correlation between thermal comfort index and physiological responses in dairy cows [24,25]. Common cow thermal comfort indices are as follows.

Temperature–humidity index [26]:(1)$$\mathrm{THI}=\left(1.8Ta+32\right)-\left(0.55-0.0055\mathrm{RH}\right)\left(1.8\mathrm{Ta}-26\right)$$

Black globe humidity index [27]:(2)BGHI=Tbg+0.36Tdp+41.5

Adjusted THI [28]:(3)$$\mathrm{AT}\mathrm{HI}=4.51+\left[0.8\mathrm{Ta}+\left(\mathrm{RH}/100\right)\left(\mathrm{Ta}-14.4\right)+46.4\right]-1.992U+0.0068\mathrm{Qsr}$$

Comprehensive climate index [29]:(4)$$\begin{array}{cc} & \mathrm{CCI}=\mathrm{eq}.\left(U\right)+\mathrm{eq}.\left(\mathrm{Qsr}\right)+\mathrm{eq}.(\mathrm{RH})\\ \mathrm{where} & \\ & \mathrm{eq}.\left(U\right)=\frac{-6.56}{\exp\left\{\frac{1}{{\left(2.26U+0.23\right)}^{0.45}}\times \left[2.9+1.14\times 10^{-6}U^{2.5}-\log_{0.3}{\left(2.26U+0.33\right)}^{-2}\right]\right\}}-0.00566U^{2}+3.33\;\mathrm{eq}.\left(\mathrm{Qsr}\right)=0.0076\mathrm{Qsr}-0.00002\mathrm{QsrTa}+0.00005{\mathrm{Ta}}^{2}\sqrt{\mathrm{Qsr}}+0.1\mathrm{Ta}-2\;\mathrm{eq}.\left(\mathrm{RH}\right)=\exp\left(0.00182\mathrm{RH}+1.8\times 10^{-5}\mathrm{Ta}\mathrm{RH}\right)\times \left(0.000054{Ta}^{2}+0.00192\mathrm{Ta}-0.0246\right)\left(\mathrm{RH}-30\right) \end{array}$$

Dairy heat load index [30]:(5)$$\mathrm{DHLI}=\frac{\left(\frac{1.681813}{1-\exp\left(-\left(-8.50749+0.206149\mathrm{Tbg}\right)+4.088399\mathrm{RH}\right)}\right)-0.0002}{1.6812-0.0002}\times 100$$

Equivalent temperature index for cattle [31]:(6)$$\mathrm{ETIC}=\mathrm{Ta}-0.0038\mathrm{Ta}\left(100-\mathrm{RH}\right)-0.1173U^{0.707}\left(39.2-\mathrm{Ta}\right)+1.86\times 10^{-4}\mathrm{TaQsr}$$

Skin temperature index for cow [32]:(7)$$\mathrm{STIC}=\frac{\left(1.73\mathrm{Ta}-\left(1-\frac{\mathrm{RH}}{100}\right)10^{\frac{7.5\mathrm{Ta}}{237+\mathrm{Ta}}}+0.116U^{0.53}\left(10Ta-300\right)+0.05\mathrm{Qsr}+16.08\right)}{\left(0.116U^{0.53}+1\right)}$$
where Ta represents air temperature (°C), Tbg represents black-globe temperature (°C), Tdp represents dew point temperature (°C), RH represents relative humidity (%), U presents wind speed (m/s), and Qsr represents solar radiation (W/m^2^).

Heat transfer from cows to the surroundings

The cow heat dissipation into the environment variable is instrumental. It quantifies the cow’s ability to shed excess heat, a function that is intricately linked to its thermal balance and overall health. These features provide a direct measure of the cow’s physiological response to environmental stimuli.

Heat loss through respiration (Qresp, W/m^2^) [33]:(8)$$Q\mathrm{resp}=F_{r}\times V_{t}\times \frac{\rho c_{p}\left(T_{\mathrm{ex}}-Ta\right)}{\left(60\times S\right)}+\rho c_{p}\left(P_{e,b}-P_{e,a}\right)/\left(\gamma r_{r}\right)$$

Heat loss through sweating (Qevap, W/m^2^) [34]:(9)$$\mathrm{Qevap}=R_{SW}\times \lambda /3600$$

Convective heat transfer from the cow’s body surface to the surroundings (Qconv, W/m^2^) [35]:(10)$$\mathrm{Qconv}=\left(T_{c}-Ta\right)\left(k_{a}\mathrm{Nu}/d\right)$$

Radiant heat transfer from the cow’s body surface to the surroundings (Qrad, W/m^2^) [35]:(11)$$\mathrm{Qrad}=\left(T_{c}-T_{\mathrm{rad}}\right)\left(4\sigma \varepsilon {\left[\left(T_{c}+T_{\mathrm{rad}}\right)/2\right]}^{3}\right)$$

The total latent heat loss (Qlat, W/m^2^):Qlat = Qresp + Qevap(12)

The total sensible heat loss (Qsens, W/m^2^):Qsens = Qconv + Qrad(13)
where S represents the body surface area (5.4 m^2^); ρc_p_ represents the volumetric specific heat capacity of air at constant pressure (1220 J·m^−3^·K^−1^); γ represents the psychrometer constant (0.066 kPa·K^−1^); λ represents the latent heat of vaporization (2260 J·g^−1^); σ represents the Steven–Boltzmann constant (5.67 × 10^−8^·W·m^−2^·K^−4^); ε represents the emissivity (0.95), d represents diameter of cows (0.8 m); k_a_ represents the thermal conductivity of air (0.025 W·m^−1^·K^−1^); and Trad represents the mean radiant temperature (≈Ta). The calculation methods for the other variables are shown in Table 2.

Average and maximum temperature of ROIs

Referring to previous studies, the regions of interest (ROI) on the cows’ bodies included the head (HD) [18], eyes (EY) [20], ears (EA) [18], faces (FA) [40], neck (NK) [38], trunk (TK) [19], udder (UD) [41], front legs (FL) [42], and hind legs (HL) [42], illustrated in Figure 3. The infrared images were analyzed using the AnalyzIR software (Fortic Inc., Jing’an, Shanghai, China), and the emissivity was set to 0.95. Within this software, different shape measurement tools were employed to delineate the ROI, and the average and maximum temperatures of all the pixels within the region were automatically calculated and obtained. The average infrared temperature (IRTave) and the maximum infrared temperature (IRTmax) are valuable parameters commonly utilized in current related research [40,43].

#### 2.1.3. Data Processing

Data processing constitutes a pivotal phase in the enhancement of data integrity and precision. It serves as a purifying mechanism, meticulously eradicating spurious noise, rectifying erroneous entries, and eliminating redundant data instances, thereby establishing a robust foundation for subsequent analytical and modeling endeavors. In this section, the data processing pipeline is delineated into three principal components.

Initially, categorical variables undergo a transformation process through encoding, thereby converting them into numerical representations that are comprehensible to the modeling algorithms. Within the dataset employed for this research, two categorical variables are identified: “time zone” and “body posture”. The variable “time zone” denotes the temporal segment of the sampling event, encompassing two distinct categories: “A.M.” and “P.M.”. These are systematically encoded as 0 and 1, respectively. Similarly, the variable “body posture” captures the postural state of the cow at the instance of sampling, with categories “standing” and “lying down” being numerically encoded as 0 and 1, respectively.

Subsequently, an assessment of missing data is conducted, as depicted in Figure A1a. A relative prevalence of missing values is discerned within the variable “milk yield” and the variables associated with infrared thermal imaging (i.e., IRTmax and IRTave). We imputed the missing values based on the median of the respective variables.

Concurrently, an outlier detection protocol is implemented to identify aberrant data points. The Z-score method, elucidated in Equation (14), is employed for this purpose, which quantifies the deviation of a data point from the mean in terms of standard deviations. Data points exhibiting a Z-score that transgresses the threshold of ±3 are classified as outliers. These outliers are replaced by the sum of the corresponding mean and triple standard deviation. The outcomes of this process are graphically represented in Figure A2b.
(14)$$z=\left[x-\mathrm{mean}\left(x\right)\right]/\mathrm{std}(x)$$
where x denotes the datapoint, and mean(x) and std(x) represent the mean and standard deviation of the variable, respectively.

Upon completion of the data processing sequence, the distribution profiles of each variable are shown in Figure A3. It is noteworthy that while the distributions of certain variables may deviate from the idealized normal distribution, the machine learning models utilized in this study exhibit a commendable degree of robustness with respect to the distributional characteristics of the input features.

### 2.2. CBT Prediction Model Development

#### 2.2.1. Machine Learning Algorithms

The machine learning algorithms selected in this study include elastic net (EN), ANN, RF, XGBoost, LightGBM, and CatBoost. The key characteristics of each algorithm are displayed in Table 3.

#### 2.2.2. Hyperparameter Optimization Algorithms

The quest for optimal hyperparameters is a critical step that significantly influences model performance. Traditional grid search, while exhaustive and straightforward, often proves to be computationally expensive and inefficient, especially when dealing with high-dimensional parameter spaces. This is where metaheuristic algorithms and Bayesian optimization emerge as superior alternatives.

Metaheuristic algorithms, such as genetic algorithms (GA), differential evolution (DE), flower pollination algorithms (FPA), particle swarm optimization (PSO), grey wolf optimization (GWO), tuna swarm optimization (TSO), slime mold algorithms (SMA), symbiotic organism search (SOS), and seagull optimization algorithm (SOA), leverage principles inspired by natural phenomena to explore the hyperparameter space more intelligently. These algorithms are designed to mimic processes such as evolution, swarm behavior, and biological interactions, enabling them to navigate complex landscapes more effectively (Table 4). By iteratively refining solutions based on feedback, they can converge on optimal hyperparameters with fewer evaluations compared to grid search. This efficiency is particularly valuable in scenarios where computational resources are limited or when the model needs to be tuned quickly.

Bayesian optimization (BO), on the other hand, employs probabilistic models to guide the search for optimal hyperparameters. By leveraging prior knowledge and updating beliefs based on observed outcomes, BO intelligently selects the next set of hyperparameters to evaluate, thereby reducing the number of trials needed. This approach is particularly well suited for optimizing hyperparameters of models with expensive-to-evaluate objective functions, such as deep neural networks.

#### 2.2.3. Model Training and Evaluation

In this study, we constructed multiple distinct feature sets based on the required features extracted from the database. These feature sets were systematically categorized into five groups: The first group includes feature sets with only environmental factors (ENV and ANM, totaling 2 feature sets; ENV represents environmental variables such as air temperature, black globe temperature, relative humidity, and solar radiation, along with time zone data used as input features. ANM incorporates all the features from the first group plus animal-related variables, i.e., milk yield, days in lactation, parity, and body posture). The second group consists of maximum infrared temperature feature sets (including IRTmax_TK, IRTmax_HL, …, IRTmax_NK, totaling 7 feature sets). The third group comprises average infrared temperature feature sets (including IRTave_TK, IRTave_HL, …, IRTave_NK, totaling 7 feature sets). The fourth group includes thermal comfort index feature sets (including THI, BGHI, …, STIC, totaling 7 feature sets). The fifth group consists of heat transfer feature sets (including Qres, Qevap, …, Qsens, totaling 6 feature sets). We assumed that animal-related variables were helpful for predicting individual BCTs in cows, so the IRTave group, IRTmax group, thermal comfort index group, and heat transfer group all included animal-related variables. Each feature set contained 3005 samples, allocated in an 80% for training and 20% for testing ratio. During training, we shuffled the sample order and employed a 5-fold cross-validation approach to enhance model performance and training efficacy.

This study took into account four evaluation metrics: the coefficient of determination (*R*^2^), the mean absolute error (MAE), the root mean square error (RMSE), and time cost. *R*^2^, typically ranging from 0 to 1, serves to gauge the goodness of fit of a model. A higher *R*^2^ value approaching 1 implies superior model performance. MAE represents the average of the absolute differences between the predicted and the actual CBT values, whereas RMSE computes the average difference between them. Lower values of MAE and RMSE signify more accurate predictions. The calculation of these three metrics is presented as follows:(15)$$R^{2}=1-\frac{\sum_{i=1}^{n}{\left(y_{t,i}-y_{p,i}\right)}^{2}}{\sum_{i=1}^{n}{\left(y_{t,i}-{\overset{-}{y}}_{t}\right)}^{2}}$$
(16)$$\mathrm{MAE}=\frac{1}{n}\sum_{i=0}^{n-1}\left|y_{t,i}-y_{p,i}\right|$$
(17)$$\mathrm{RMSE}=\sqrt{\frac{1}{n}\sum_{i=0}^{n-1}{\left(y_{t,i}-y_{p,i}\right)}^{2}}$$
where *n* represents the number of samples, *y_t,i_* and *y_p,i_* represent the actual CBT and the predicted CBT of the *i*th sample, respectively, and ${\overset{-}{y}}_{t}$ denotes the average of the actual CBTs.

Within the context of this study, “time cost” specifically quantifies the duration required to discover the most effective hyperparameters via optimization algorithms. The efficiency of these algorithms is inversely proportional to the time consumed; hence, a shorter time to convergence signifies superior performance. To ensure a level playing field for comparative analyses, all optimization algorithms were executed under identical experimental settings. Among them, the population and epochs were set to 30 and 200, respectively.

The experimentations were run on the operating system of Windows 11 Professional Workstation Edition 23H2 with 32 GB RAM and CPU of Intel (R) Core (TM) i7-11800H (2.30 GHz). The algorithms for comparison were coded by Python 3.9.

### 2.3. Feature Analysis Based on SHAP

Machine learning models frequently operate as black boxes, presenting interpretability challenges for researchers attempting to understand model predictions. This opacity often leads dairy farm producers, operators, or managers to be hesitant about adopting solutions based on such models. The traditional game-theoretic concept of the Shapley value provides a valuable method for elucidating the impact of individual input features on model outcomes [58]. This impact is assessed through the feature’s marginal contribution, quantified by the Shapley value [58,59]. Equation (4) defines the computation of the Shapley value ∅_i_ [58]:(18)$$\emptyset_{i}=\sum_{S\subseteq F\backslash \left\{i\right\}}\frac{\left|S\right|!\left(\left|F\right|-\left|S\right|-1\right)}{\left|F\right|!}\left[f_{S\cup \left\{i\right\}}\left(x_{S\cup \left\{i\right\}}\right)-f_{S}\left(x_{S}\right)\right]$$
where F denotes the complete set of features, S signifies the subset of features not including the ith feature, and $f_{S\cup \left\{i\right\}}\left(x_{S\cup \left\{i\right\}}\right)-f_{S}\left(x_{S}\right)$ denotes the variation in predictions with and without the ith feature. Since the effect of omitting a feature is contingent on other features, the variations are computed across all potential subsets S ⊆ F\{i}. The Shapley value is then derived by averaging these variations, weighted by their occurrence probabilities [8,54]. 

The Shapley value offers only local explanations and demands substantial computational resources [60]. In response, Lundberg and Lee developed SHAP in 2017, which not only retains the benefits of the Shapley value but also markedly enhances computational efficiency [60]. Notably, SHAP transforms the computational complexity for tree-based machine learning models from exponential to polynomial [60]. As depicted in Figure 4 (adapted from Lundberg et al. [60]), SHAP TreeExplainer facilitates local explanations to clarify model prediction dynamics. This computational enhancement enables SHAP to aggregate local explanations across the dataset, thereby offering global insights while preserving local fidelity to the model [60]. SHAP employs the SHAP value as a consistent metric for feature importance, equivalent to the Shapley value of a conditional expectation function derived from the original model [58]. This research utilized SHAP to analyze each optimal CBT prediction model, identifying pivotal features affecting CBT and offering comprehensive insights into the mechanisms influencing CBT from both global and local viewpoints.

## 3. Results

### 3.1. Model Performance under Different Feature Sets

#### 3.1.1. Impact of Adding Animal-Related Features on Model Performance

Figure 5 reveals the significant enhancement in predictive capability when animal-related variables were included. Building upon the ENV feature set, which consists solely of environmental variables (Ta, Tbg, RH, U, Qsr) and time zone (TZ), we further incorporated milk yield, days of lactation, parity, and body posture to create the ANM feature set. The results showed that models trained with the ANM feature set exhibited a significantly higher R^2^ value (paired sample t-statistic = −6.664, df = 5, *p* = 0.001), as well as lower MAE and RMSE (t-statistic for MAE = 5.007, df = 5, *p* = 0.004; RMSE t-statistic = 6.5428, df = 5, *p* = 0.001). This underscores the importance of individual animal characteristics in improving prediction accuracy.

#### 3.1.2. Changes in Model Performance under IRTmax Feature Sets

Figure 6 illustrates the changes in model performance when combining different maximum infrared temperatures (IRTmax) with animal-related variables. Experiments indicate that models trained using the IRTmax_TK feature set performed optimally on the test dataset, characterized by the highest R^2^ value (0.510 ± 0.015), lowest MAE (0.241 ± 0.004 °C), and RMSE (0.304 ± 0.005 °C).

#### 3.1.3. Changes in Model Performance under IRTave Feature Sets

As shown in Figure 7, when different average infrared temperatures (IRTave) combined with animal-related variables were used as input features, the IRTave_TK feature set enabled the model to achieve optimal performance, reflected in the highest R^2^ value (0.516 ± 0.021), lowest MAE (0.239 ± 0.005 °C), and RMSE (0.302 ± 0.007 °C). In conjunction with the analysis in Section 3.1.2, we can infer that the infrared temperature, particularly its average, at the trunk is a preferred predictor for predicting CBT in dairy cows.

#### 3.1.4. Changes in Model Performance under Thermal Comfort Index Feature Sets

Figure 8 demonstrates the predictive efficacy of models when thermal comfort indices were used in conjunction with animal-related variables. The results indicated that when thermal comfort indices served as input features, the *R*^2^ values for CBT prediction models ranged from 0.419 to 0.500, MAE fluctuated between 0.246 and 0.268 °C, and RMSE varied from 0.307 to 0.331 °C. Specifically, when STIC is selected as the thermal comfort index, the model exhibits better predictive performance.

#### 3.1.5. Changes in Model Performance under Heat Transfer Feature Sets

Figure 9 shows the performance changes of models when heat transfer variables were considered alongside animal-related variables. Except for Qres and Qconv, CBT prediction models trained with other heat transfer variables exhibited similar performances, with R^2^ values concentrated between 0.484 and 0.498, MAE maintained within 0.247 to 0.249 °C, and RMSE varying from 0.308 to 0.312 °C.

### 3.2. Model Evaluation under Different Optimization Algorithms

#### 3.2.1. Time Cost of Hyperparameter Optimization under Different Algorithms

Figure 10 contrasts the time consumed by different optimization algorithms in the quest for the optimal CBT prediction model. For models such as ANN, RF, XGBoost, and LightGBM, Bayesian optimization (BO) was the most efficient method for hyperparameter tuning, while symbiotic organisms search (SOS) was the most time-consuming. For elastic net (EN)-based models, BO and SOS had similar time costs, both relatively high. In CatBoost-based models, the slime mold algorithm (SMA) was the quickest optimization strategy, whereas the seagull optimization algorithm (SOA) was the most time-intensive.

#### 3.2.2. Comprehensive Comparison of Optimized CBT Prediction Models

Figure 11 summarizes the evaluation metrics for all combinations of optimization algorithms and machine learning algorithms applied to CBT prediction models. In terms of time efficiency, differential evolution (DE)-optimized elastic net (DE-EN) performed best, with a runtime of 2.41 s, which is approximately 1/4500th of the time required by SOS-optimized artificial neural network (SOS-ANN) at 10,957.58 s (Figure 11a). Due to the limited number of hyperparameters in elastic net, all EN-based models showed shorter optimization times. In terms of predictive accuracy (Figure 11b–d), the symbiotic search algorithm-optimized CatBoost model (SOS–CatBoost) achieved the best R^2^ value (0.540) and RMSE (0.294 °C). The grey wolf optimization (GWO)-optimized XGBoost model (GWO–XGBoost) excelled in MAE (0.232). Considering R^2^, RMSE, MAE, and optimization time collectively, the GWO–XGBoost model achieved the best balance between predictive precision and computational efficiency, making it the overall best-performing CBT prediction model.

### 3.3. SHAP Analysis

#### 3.3.1. Feature Importance

Figure 12 exhibits the SHAP summary plot derived from the GWOXGBoost model for predicting cow CBT. This plot offers profound insights into how critical features contribute to the prediction of CBT, alongside illustrating their relative importance. The average absolute SHAP values in Figure 12a depict the mean impact magnitude on the model’s output. Among these, the average trunk infrared temperature (IRTave_TK) emerges as the paramount feature, boasting the highest SHAP importance value of 0.23. It is followed by the time zone (TZ), with a value of 0.05, and then days in lactation (DOL), at 0.03. In Figure 12b, the X-axis delineates the SHAP values, which reflect the effect of each feature on the CBT prediction. A positive SHAP value signifies a positive contribution, thus increasing the CBT, while a negative value indicates a decrease in CBT. Each point represents an individual sample, with blue points indicating lower feature values and red points denoting higher ones. For numerical variables, a notable correlation exists between IRTave_TK values and SHAP values, whereas DOL and parity (PA) display an inverse relationship.

#### 3.3.2. Single-Feature Analysis and Feature Interaction Analysis

Figure 13 shows the SHAP dependence plots for the GWO–XGBoost model’s CBT predictions. These plots elucidate how CBT predictions vary with changes in feature values and whether this variation is influenced by interactions with other features. Focusing on the top four most influential features, the SHAP dependence plots reveal how feature values on the X-axis correlate with the SHAP values for CBT prediction on the Y-axis for each cow sample. For instance, Figure 13a illustrates that when IRTave_TK exceeds 34 °C, the SHAP value surpasses 0.00, significantly elevating CBT values up to 0.55 °C. Figure 13c suggests a decline in CBT values once the DOL exceeds 200 days. Figure 13b,d indicate that cows lying down during the P.M. experience higher CBT values.

To explore how the influence of key features on CBT interacts with others, we initially generated dependency plots for all possible pairs among the top four features. From these, we selected those demonstrating significant and meaningful interactions, exemplified in the final paired dependency plots shown in Figure 13e–h. Here, the gradient color of the points represents variations in the second feature’s value. By observing the pronounced dispersion of red and blue, one can ascertain how the effect of a primary feature on CBT is modulated by another. Figure 13e,h highlighted that the impact of IRTave_TK on CBT varies according to body posture; specifically, standing cows exhibit a more pronounced increase in CBT when IRTave_TK is above 34 °C compared to cows in a reclining position. Figure 13f demonstrates that high IRTave_TK values lead to a quicker rise in CBT during the P.M. period, while low IRTave_TK values result in a slower decline in CBT during the A.M. hours. In Figure 13g, high DOL values consistently coincide with low milk yield (MY) values, leading to a reduction in CBT.

#### 3.3.3. Local Interpretation

Figure 14 presents the SHAP waterfall plots, which provide a detailed explanation of the local predictions made by the GWO–XGBoost model. The waterfall plots sequentially illustrate the contribution of each feature to the CBT prediction for individual instances, declining in order of impact, which may not align with the overall results depicted in the bee swarm plot. Contributions to CBT predictions are indicated by blue and red bars corresponding to their magnitude (blue = negative, red = positive). The feature values for each instance are represented by numbers to the left of each factor within the waterfall plot. The model’s prediction is displayed at the upper left corner of each waterfall plot as f(x), with E[f(X)] representing the average prediction across the dataset at the bottom. In six randomly selected instances (Figure 14a–f), it becomes evident that IRTave_TK and TZ rank as the two most prominent features in four cases (Figure 14c–f), in agreement with the summary plot. However, there is inconsistency in the order of significance for other features in local predictions, potentially due to minor differences in their overall feature significance. In Figure 14a,b, TZ takes precedence over IRTave_TK, suggesting that under conditions where actual CBT is relatively low, perhaps below 39.0 °C, TZ assumes greater predictive importance than IRTave_TK.

## 4. Discussion

### 4.1. Feature Sets

Traditionally, models predicting the core body temperature (CBT) of dairy cows have tended to rely heavily on environmental parameters as primary independent variables [10]. However, such models struggle to explain the subtle differences in CBT among individuals under identical environmental conditions. To achieve more precise individual CBT predictions, our study introduced variables related to the physiological and behavioral traits of the cow. This step was validated by experimental results showing that models incorporating animal-specific parameters exhibited higher predictive accuracy than those based solely on environmental conditions. Nevertheless, there remains a lack of clear guidelines specifying which particular factors are most suitable as inputs for CBT predictive models.

Previously, Gorczyca et al. [16] adopted environmental temperature, relative humidity, wind speed, and solar radiation as inputs, achieving an artificial neural network (ANN) prediction performance with R^2^ = 0.472, MAE = 0.351 °C, RMSE = 0.434 °C. Shu et al. [15], on the other hand, included milk yield over the previous three days, calving season, birth season, age in months, lactation day, parity, body posture, and time zone in their model, which, after adjustment via grid search, resulted in an ANN model with R^2^ = 0.45, RMSE = 0.31 °C. While these studies share similar data environments and sets with ours, our innovation lies in the introduction of thermal comfort indices, heat transfer variables, and infrared temperature information of the cow’s surface, aimed at enhancing the predictive power of the models.

Although machine learning models can handle a multitude of features, an excess of them can lead to overly complex models and diminished generalization capability, and may introduce noise and redundant information. Given the high correlation among newly created variables, using them collectively as model inputs can trigger multicollinearity issues; even if the model’s predictive performance is acceptable, it may reduce interpretability [61]. To address this, we subdivided the dataset into distinct feature subsets and evaluated their utility individually. Overall, the feature set based on average infrared temperature (IRTave) surpassed those of thermal comfort and heat transfer variables, and outperformed the set based on maximum infrared temperature (IRTmax). IRTmax, as a temperature of a single pixel, has limited representational breadth, whereas IRTave provides a more comprehensive reflection of regional temperatures, responding more stably to external fluctuations [40]. The reason why trunk mean infrared temperature performs better as a predictive factor is not clearly answered in the existing literature, possibly because the trunk area is larger, encompassing more pixels, and is closely associated with rumen temperature.

### 4.2. Optimization Algorithms

In the process of constructing CBT predictive models, we employed various optimization algorithms to fine-tune the hyperparameters of machine learning models. Among these, the CatBoost model optimized with symbiotic organisms search (SOS) showed the best performance, albeit with tuning times exceeding 110,950 s. By contrast, the XGBoost model optimized with the grey wolf optimizer (GWO) achieved slightly inferior predictive accuracy (R^2^ = 0.539, MAE = 0.232 °C, RMSE = 0.295 °C) but required only about 140 s for tuning, demonstrating a better balance between efficiency and performance. The GWO algorithm’s equilibrium between exploration and exploitation, through simulating the hunting behavior of grey wolf packs, enables a compromise between global search and local optimization, contributing to improved predictive accuracy [52].

Bayesian optimization (BO) demonstrated the shortest optimization time in four of the six machine learning models we evaluated. This stems from BO’s ability to effectively leverage past evaluation results to guide subsequent parameter selection, reducing unnecessary iterations. By constructing a Gaussian process model to approximate the objective function and applying an acquisition function to determine the next assessment point, BO avoids blind searches across the entire space [57]. In our study, where each algorithm had no more than five hyperparameters to optimize, Bayesian optimization displayed its unique advantages. However, this conclusion is limited by the current dataset and feature combination; under different scenarios, such as varying data compositions, distributions, or hyperparameter settings, the performance of optimization algorithms may differ.

### 4.3. Feature Analysis

We utilized SHAP values as a tool to help decipher complex machine learning models, making them more interpretable. In the GWO-XGBoost model, IRTave_TK was identified as the feature with the greatest impact on model output. This finding makes logical sense since, among all predictive variables, IRTave_TK is the sole temperature–type variable, and there is a strong positive correlation between cow surface temperature and core temperature. Following IRTave_TK, the next significant features were time zone (TZ), days in lactation (DOL), and body posture (BP). The time zone is the second most influential factor, closely tied to environmental temperature and the periodic changes in heat accumulation within the cow. Typically, daily temperature peaks occur in the late afternoon or evening, mirroring the cow’s cycle of heat accumulation during the day and dissipation at night, suggesting that mornings may be periods of lower heat accumulation in cows [2,62]. Lactation day and body posture are important because they directly relate to the cow’s heat production and dissipation. After the peak lactation period of around 60 to 100 days, an increase in lactation days coincides with a decrease in milk production, thus reducing metabolic heat production and tempering the rise in core temperature [63,64]. Milk yield itself contributes less to model output, perhaps because lactation day already encapsulates much of this information. Body posture affects the cow’s heat dissipation efficiency [65]; Wang et al. [66] found that when cows are lying down, approximately 16% less body surface area is available for cooling.

### 4.4. Limitations and Prospects

The limitations of our study primarily pertain to the scale and source of the dataset, which may constrain the predictive capabilities of the models. Future research could consider expanding the dataset or integrating more diverse data sources to further refine the models. In terms of model improvement, exploring stacking and other ensemble methods to combine predictions from multiple models could significantly enhance predictive accuracy. Considering the time series nature of CBT, employing architectures such as long short-term memory networks (LSTMs) or gated recurrent units (GRUs) in deep learning frameworks might unlock new predictive potential. On the front of hyperparameter optimization, while we have already applied various heuristic and Bayesian optimization algorithms, given the rapid advancements in the field, experimenting with more novel optimization algorithms could further improve model precision and efficiency.

## 5. Conclusions

This study employed machine learning techniques, through the comprehensive assessment of multiple feature sets, to develop an effective core body temperature (CBT) prediction model for dairy cows. Analysis showed that the predictive ability of the model significantly improved when combined with specific animal-related features and infrared temperature measurements. Particularly, the feature set incorporating the average infrared temperature of the trunk (IRTave_TK) demonstrated the highest predictive accuracy, highlighting the pivotal role of infrared temperature in monitoring heat stress in dairy cows. Moreover, the GWO–XGBoost model was identified as the optimal CBT prediction model due to its outstanding predictive accuracy and efficient computational performance. Through SHAP analysis, critical factors influencing CBT and their interactions were revealed, providing important insights into the mechanisms of heat stress and informing the design of targeted intervention strategies. The outcomes of the study hold potential for advancing precision livestock management, enhancing animal welfare, and boosting farm economic efficiency.

## Figures and Tables

**Figure 1 animals-14-02724-f001:**
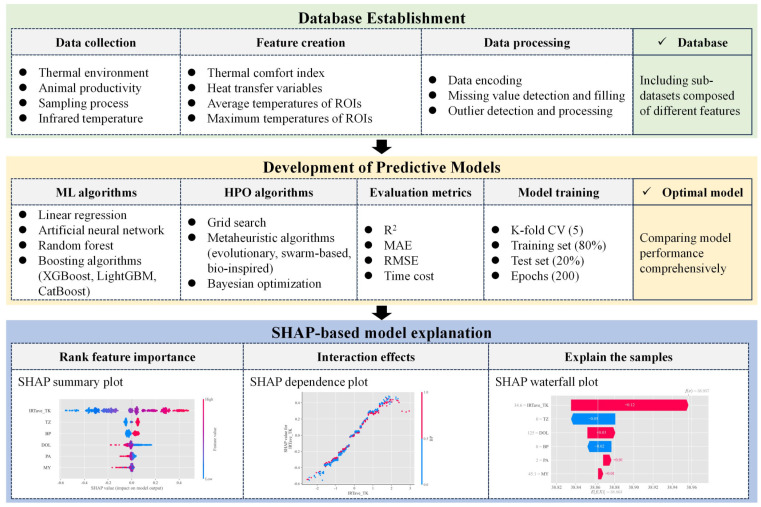
Research framework of this study.

**Figure 2 animals-14-02724-f002:**
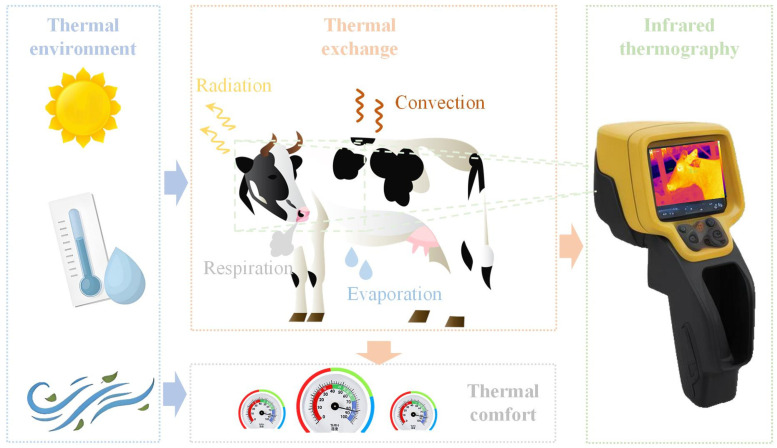
Illustration of valuable predictor associated with CBT of cows.

**Figure 3 animals-14-02724-f003:**
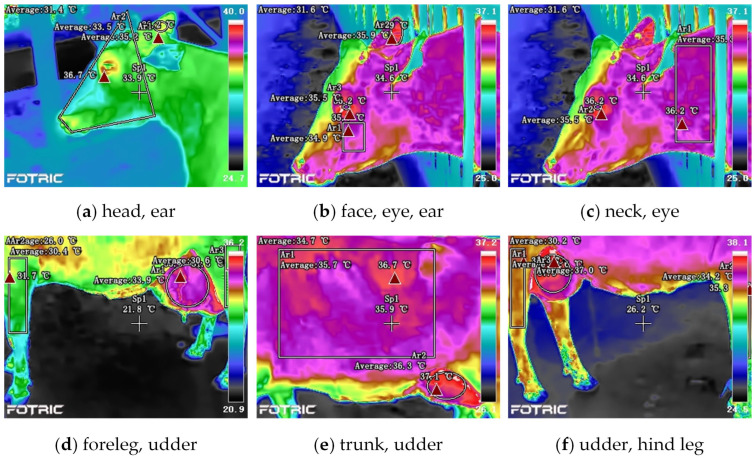
Regions of interest (ROI) on the body surface of dairy cows and measurement tools to contour ROIs.

**Figure 4 animals-14-02724-f004:**
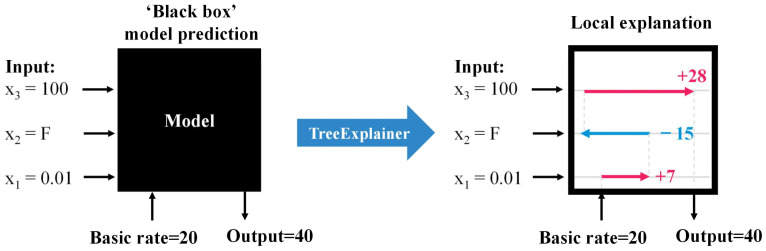
Illustration of SHAP local explanations, adapted from Lundberg et al. [60]. The values and variables depicted are for illustrative use only and do not reflect real data.

**Figure 5 animals-14-02724-f005:**
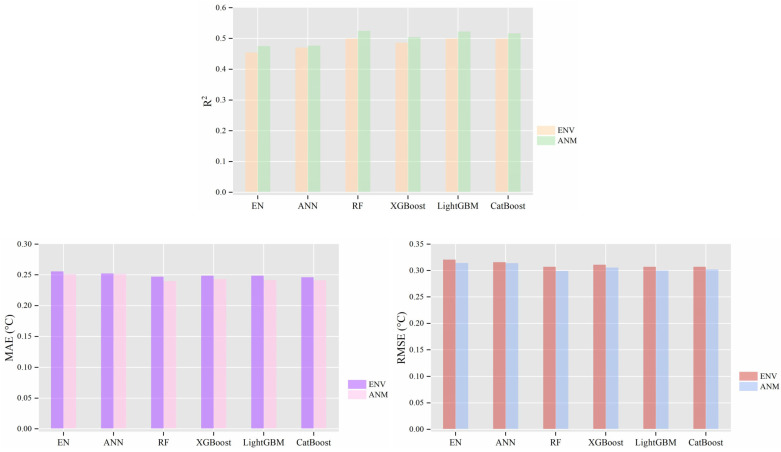
Performance of the dairy cow core temperature prediction model based on machine learning algorithms on the test set. ENV indicates the training set where the predictive variables were only air temperature, black globe temperature, relative humidity, solar radiation, and time zone. ANM, in addition to the above, included milk yield, days of lactation, parity, and body posture variables in the dataset. EN stands for elastic net, ANN for artificial neural network, RF for random forest, XGBoost for extreme gradient boosting, LightGBM for light gradient boosting machine, and CatBoost for categorical boosting. *R*^2^ represents the coefficient of determination, MAE the mean absolute error, and RMSE the root mean square error.

**Figure 6 animals-14-02724-f006:**
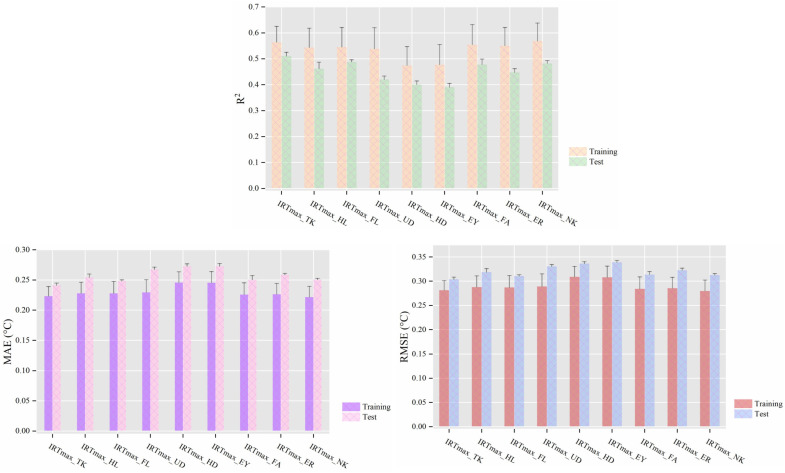
Mean performance (±standard deviation, N = 6) of cow CBT prediction models when the maximum infrared temperatures (IRTmax) from different body regions were used as input features. Note: all feature sets included cow-related variables (milk yield, days of lactation, parity, and body posture). TK stands for trunk, HL for hind legs, FL for front legs, UD for udder, HD for head, EY for eye region, FA for face, ER for ears, and NK for neck. ‘Training’ and ‘Test’ refer to performance metrics obtained on the training and testing sets, respectively. *R*^2^ represents the coefficient of determination, MAE the mean absolute error, and RMSE the root mean square error.

**Figure 7 animals-14-02724-f007:**
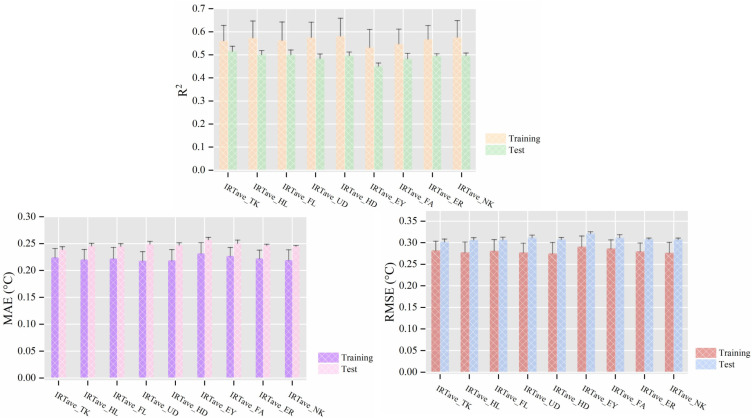
Mean performance (±standard deviation, N = 6) of cow CBT prediction models when the average infrared temperatures (IRTave) from different body regions were used as input features. Note: All feature sets included cow-related variables (milk yield, days of lactation, parity, and body posture). TK stands for trunk, HL for hind legs, FL for front legs, UD for udder, HD for head, EY for eye region, FA for face, ER for ears, and NK for neck. ‘Training’ and ‘Test’ refer to performance metrics obtained on the training and testing sets, respectively. *R*^2^ represents the coefficient of determination, MAE the mean absolute error, and RMSE the root mean square error.

**Figure 8 animals-14-02724-f008:**
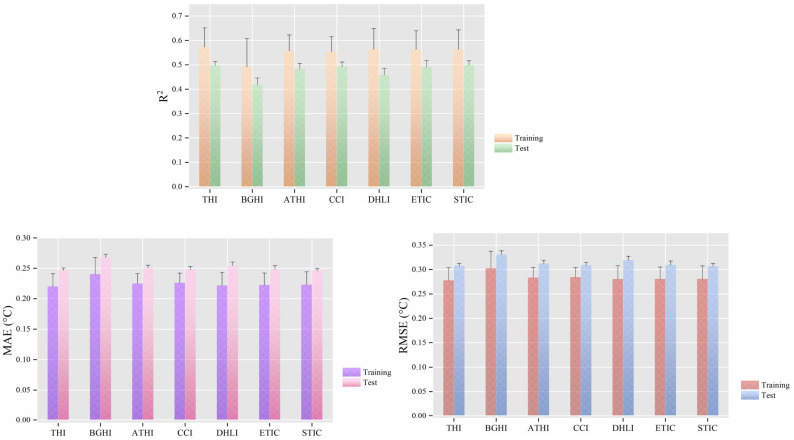
Mean performance (±standard deviation, N = 6) of cow CBT prediction models when the thermal comfort indices were used as input features. Note: all feature sets included cow-related variables (milk yield, days of lactation, parity, and body posture). THI is the temperature-humidity index, BGHI is the black globe humidity index, ATHI is the adjusted THI, CCI is the comprehensive climatic index, DHLI is the heat load index for dairy cows, ETIC is the equivalent temperature index for cows, and STIC is the skin temperature index for cows. ‘Training’ and ‘Test’ refer to performance metrics obtained on the training and testing sets, respectively. *R*^2^ represents the coefficient of determination, MAE the mean absolute error, and RMSE the root mean square error.

**Figure 9 animals-14-02724-f009:**
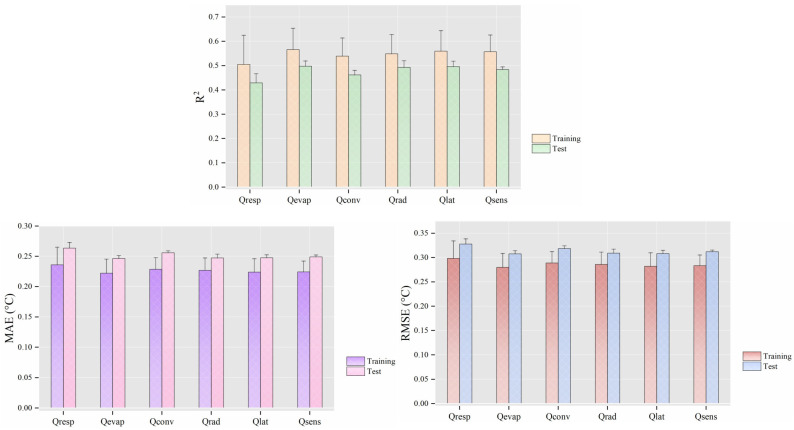
Mean performance (±standard deviation, N = 6) of cow CBT prediction models when the heat transfer variables were used as input features. Note: all feature sets included cow-related variables (milk yield, days of lactation, parity, and body posture). Qresp stands for the respiratory heat loss, Qevap for the evaporative heat loss from the skin surface, Qconv for the convective heat flux, Qrad for the radiative heat flux, Qlat for the total latent heat loss, and Qsens for the sensible heat flux. ‘Training’ and ‘Test’ refer to performance metrics obtained on the training and testing sets, respectively. *R*^2^ represents the coefficient of determination, MAE the mean absolute error, and RMSE the root mean square error.

**Figure 10 animals-14-02724-f010:**
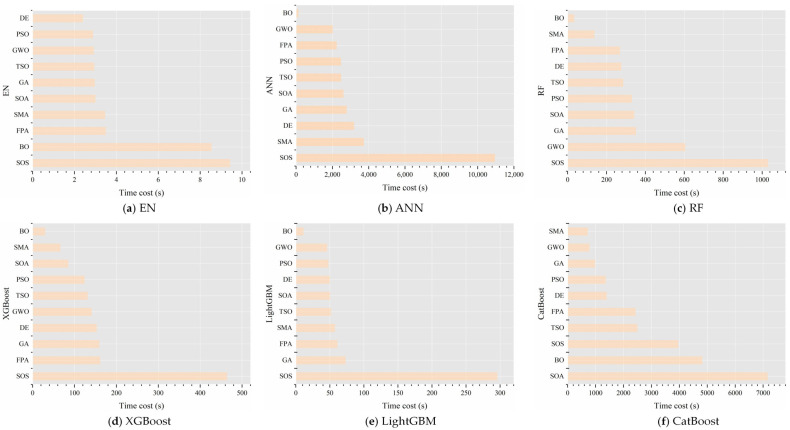
Time cost for obtaining the best CBT prediction models by different machine learning algorithms under various optimization algorithms. EN stands for elastic net, ANN for artificial neural network, RF for random forest, XGBoost for extreme gradient boosting, LightGBM for light gradient boosting machine, and CatBoost for categorical boosting. GA represents genetic algorithm, DE stands for differential evolution, FPA for flower pollination algorithm, PSO for particle swarm optimization, GWO for grey wolf optimizer, TSO for tuna swarm optimization, SMA for slime mold algorithm, SOS for symbiotic organisms search, SOA for seagull optimization algorithm, and BO for Bayesian optimization.

**Figure 11 animals-14-02724-f011:**
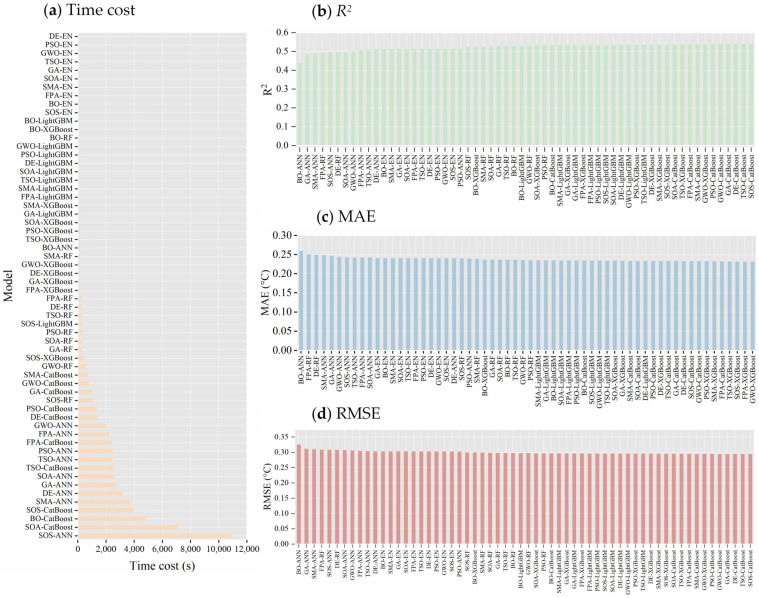
Evaluation metrics for machine learning-based CBT prediction models optimized by optimization algorithms. EN stands for elastic net, ANN for artificial neural network, RF for random forest, XGBoost for extreme gradient boosting, LightGBM for light gradient boosting machine, and CatBoost for categorical boosting. GA represents genetic algorithm, DE stands for differential evolution, FPA for flower pollination algorithm, PSO for particle swarm optimization, GWO for grey wolf optimizer, TSO for tuna swarm optimization, SMA for slime mold algorithm, SOS for symbiotic organisms search, SOA for seagull optimization algorithm, and BO for Bayesian optimization. R^2^ denotes the coefficient of determination, MAE represents the mean absolute error, and RMSE stands for the root mean square error.

**Figure 12 animals-14-02724-f012:**
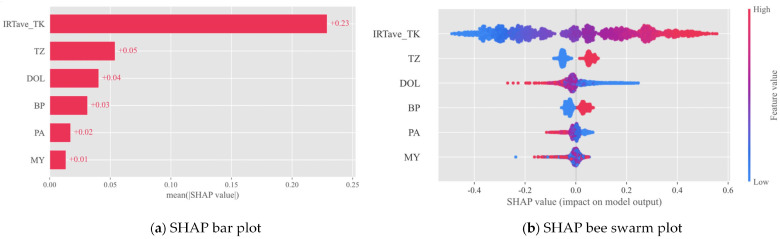
SHAP summary plot for identifying key features affecting CBT prediction. IRTave_TK stands for the average infrared temperature at the trunk, TZ for time zone, DOL for days of lactation, BP for body posture, PA for parity, and MY for milk yield.

**Figure 13 animals-14-02724-f013:**
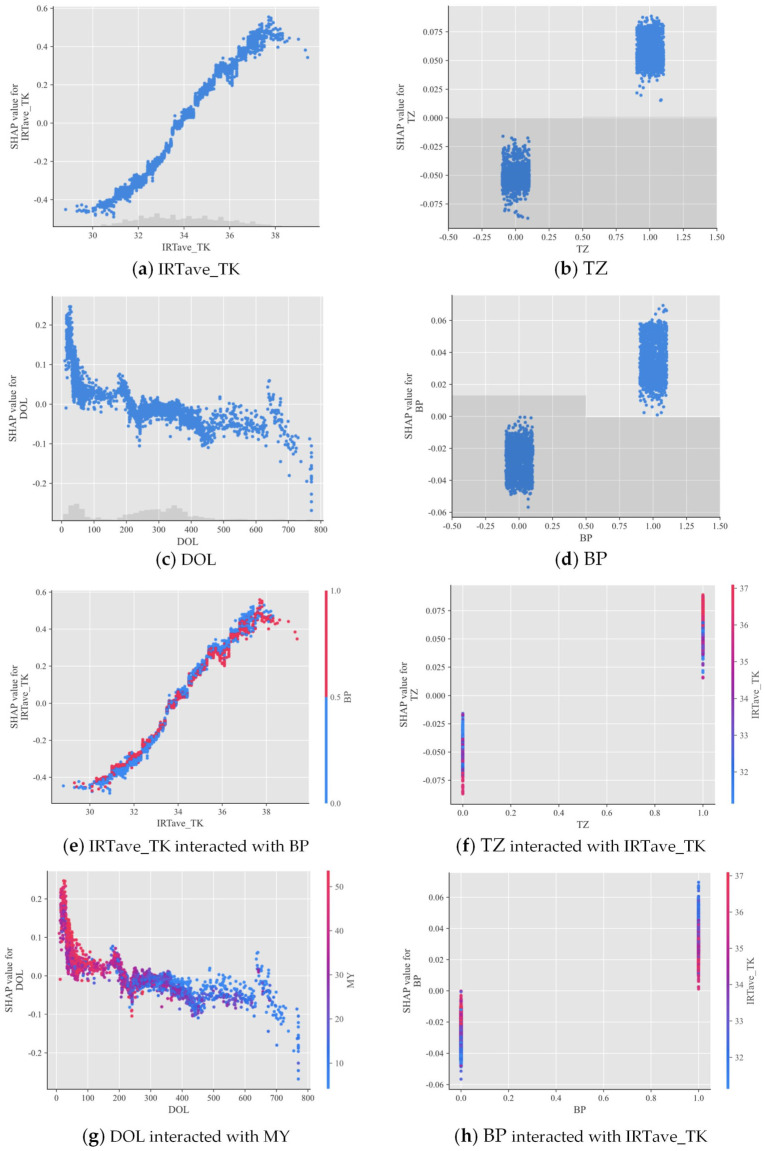
SHAP dependence plot for identifying how key features impact CBT and how this impact mechanism interacts with another feature. IRTave_TK stands for the average infrared temperature at the trunk, TZ for time zone, DOL for days of lactation, BP for body posture, PA for parity, and MY for milk yield.

**Figure 14 animals-14-02724-f014:**
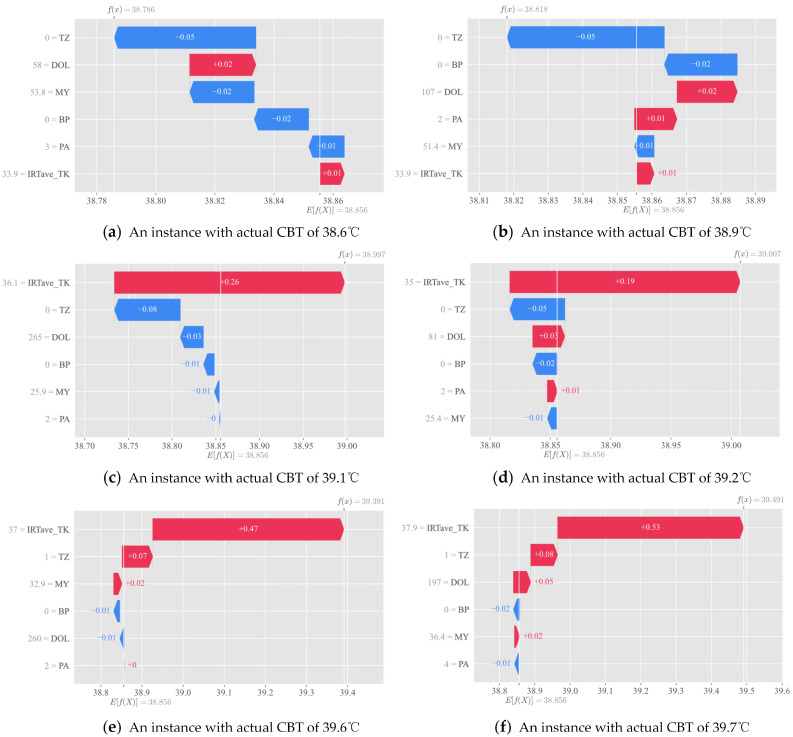
SHAP waterfalls: examples of individual effect of features on CBT. IRTave_TK stands for the average infrared temperature at the trunk, TZ for time zone, DOL for days of lactation, BP for body posture, PA for parity, and MY for milk yield.

**Table 1 animals-14-02724-t001:** The application of machine learning methods in the prediction of core body temperature of dairy cows in recent years.

Year	Data Source	Predictors	ML Algorithms	Evaluation Metrics	Hyperparameter Tune	Optimal Model	Ref.
2023	Karnobat, Bulgaria	Temperature–humidity index, respiratory frequency, pulse, time of measurement	WeightedEnsemble_L2, NeuralNetTorch, XGBoost, CatBoost	*R*^2^, MAE, MedAE, RMSE	N/A ^1^	WeightedEnsemble_L2	[14]
2023	Shandong, China	Air temperature, relative humidity, wind speed, time block, posture, age in months, parity, days in milk, calving season, birth season, DMY1D ^2^, DMY2D, DMY3D	Regularized linear regression, random forest, gradient boosted machine, artificial neural network	*R*^2^, RMSE	Grid search	artificial neural network	[15]
2020	California, U.S.	Air temperature, relative humidity, wind speed, solar radiation	Penalized linear regression, random forest, gradient boosted machine, artificial neural network	*R*^2^, MAE, RMSE	Random search	artificial neural network	[16]
2014	Minas Gerais, Brazil	Dry-bulb temperature, relative humidity	Linear regression, artificial neural network, neuro-fuzzy network	*R*^2^, RMSE	N/A	artificial neural network	[17]

^1^ N/A = not available. ^2^ DMY1D = daily milk yield on the day before the test day (kg/day); DMY2D = daily milk yield on the 2nd day before the test day (kg/day); DMY3D = daily milk yield on the 3rd day before the test day (kg/day).

**Table 2 animals-14-02724-t002:** Variables related to heat exchange of dairy cows with the environment and their calculation methods.

Variable	Computation Method	Ref.
Skin temperature (T_s_, °C)	$T_{s}=\frac{0.173\mathrm{Ta}-\left(0.1-\frac{\mathrm{RH}}{1000}\right)\times 10^{\frac{7.5\mathrm{Ta}}{237.5+\mathrm{Ta}}}+0.116U^{0.53}\mathrm{Ta}+\frac{5\mathrm{Qsr}}{1000}+31.608}{0.116U^{0.53}+1}$	[32]
Coat temperature (T_c_, °C)	$T_{c}=18.76+0.908\mathrm{Ta}-0.011{\mathrm{Ta}}^{2}$	[36]
Exhaled air temperature (T_ex_, °C)	$T_{\mathrm{ex}}=17.0+0.3Ta+\exp(0.01611\mathrm{RH}+0.0387Ta)$	[37]
Sweating rate (R_sw_, g/(m^2^·h))	$R_{\mathrm{sw}}=\left\{\begin{array}{c} 1.1665{T_{s}}^{2}-64.166T_{s}+894.35,T_{s}<35\\ 4.2976T_{s}-71.289,T_{s}\geq 35 \end{array}\right.$	[34]
Saturation vapor pressure at Ta (P_e,a_, kPa)	$P_{e,a}=0.611\times 10^{\frac{7.5\mathrm{Ta}}{\left(\mathrm{Ta}+237.3\right)}}$	[38]
Respiratory frequency (F_r_, breaths per min)	$F_{r}=\exp\left(2.966+0.0218Ta+0.00069Ta\right)$	[37]
Tidal volume (V_t_, m^3^)	$V_{t}=0.0189{F_{r}}^{-0.463}$	[37]
Respiratory evaporative resistance (r_r_, s/m)	$r_{r}=\frac{100}{2.7\times 10^{-4}F_{r}+5\times 10^{-3}}$	[39]

Note: Ta represents air temperature (°C), RH represents relative humidity (%), U represents wind speed (m/s), and Qsr represents solar radiation (W/m^2^).

**Table 3 animals-14-02724-t003:** A brief overview of the machine learning algorithms and their hyperparameter spaces.

Algorithms		Hyperparameters		
	Overview		Description	Range
EN	EN combines the penalties of ridge regression and lasso, allowing it to select features while also handling correlated predictors [15].	‘alpha’	Constant that multiplies the penalty terms	[0.0001, 10]
		‘l1_ratio’	Ratio of L1 penalty	[0.0001, 1.0]
ANN	ANN are highly adaptable, capable of processing complex data patterns, and inspired by the biological neural networks found in the human brain [15].	‘max_iter’	Maximum number of iterations	[200, 1000]
		‘learning_rate_init’	The initial learning rate	[0.001, 0.1]
		‘alpha’	Strength of the L2 regularization term	[0.0001, 10]
		‘activation’	Activation function	[‘relu’, ‘tanh’]
		‘hidden_layer_sizes’	Number of neurons in the ith hidden layer	[(100), (100, 50)]
RF	RF is an ensemble method that uses multiple decision trees to improve prediction accuracy and reduce overfitting. It can capture complex relationships in the data by averaging the predictions of many trees [44].	‘n_estimators’	Number of trees	[100, 1000]
		‘max_depth’	Maximum depth of the tree	[3, 10]
		‘min_samples_split’	Minimum number of samples for each split	[0.002, 0.2]
		‘min_samples_leaf’	Minimum number of samples for each node	[0.001, 0.1]
		‘max_features’	Features at each split	[‘sqrt’, ‘log2’]
XGBoost	XGBoost is a gradient-boosting algorithm that sequentially adds weak learners (typically decision trees) to the model, optimizing the addition of each tree to minimize prediction errors. It handles complex relationships within the data effectively [45].	‘n_estimators’	Number of trees	[100, 1000]
		‘learning_rate’	Boosting learning rate	[0.01, 0.2]
		‘subsample’	Subsample ratio of the training instance	[0.1, 1.0]
		‘reg_alpha’	L1 regularization term	[0.01, 10.0]
		‘reg_lambda’	L2 regularization term	[0.01, 10.0]
LightGBM	LightGBM is a gradient-boosting framework that uses histogram-based techniques to construct decision trees more efficiently. This results in faster training times, better scalability, and improved accuracy on large datasets [46].	‘num_iteration’	Number of iterations	[100, 1000]
		‘learning_rate’	Learning rate	[0.01, 0.2]
		‘bagging_fraction’	Sample rate for bagging	[0.1, 1.0]
		‘lambda_l1’	L1 regularization term	[0.01, 10.0]
		‘lambda_l2’	L2 regularization term	[0.01, 10.0]
CatBoost	CatBoost is an advanced boosting algorithm specifically designed to handle categorical features effectively. It uses innovative techniques such as ordered boosting and target statistics to improve predictive accuracy in datasets with categorical variables [47].	‘iterations’	Number of iterations	[100, 1000]
		‘learning_rate’	Learning rate	[0.01, 0.2]
		‘l2_leaf_reg’	L2 regularization term	[0.01, 10.0]
		‘subsample’	Sample rate for bagging	[0.1, 1.0]
		‘depth’	Depth of the trees	[3, 10]

**Table 4 animals-14-02724-t004:** Basic characteristics of the hyperparameter optimization algorithms used in the study.

Optimization Method	Category	Overview	Suitability	Ref.
Genetic Algorithm	Evolutionary	Mimics natural selection and genetics; uses selection, crossover, mutation	Complex, non-linear optimization problems	[48]
Differential Evolution	Evolutionary	Population-based stochastic algorithm; uses differences in parameter vectors	Global optimization in continuous search spaces	[49]
Flower Pollination Algorithm	Nature-inspired	Inspired by flower pollination behavior; uses global and local pollination strategies	Continuous optimization problems	[50]
Particle Swarm Optimization	Swarm-based	Mimics social behavior of bird flocking; adjusts positions based on personal and global bests	Continuous optimization problems	[51]
Grey Wolf Optimizer	Nature-inspired	Mimics the hunting mechanism of grey wolves; uses encircling, hunting, and attacking phases	Continuous and discrete optimization problems	[52]
Tuna Swarm Optimization	Swarm-based	Inspired by tuna migration behavior; combines exploration and exploitation strategies	Continuous optimization problems	[53]
Slime Mold Algorithm	Nature-inspired	Mimics the foraging behavior of slime mold; uses contraction and expansion mechanisms	Pathfinding and optimization problems	[54]
Symbiotic Organisms Search	Nature-inspired	Inspired by mutualistic symbiotic relationships; uses mutualism, commensalism, and parasitism strategies	Continuous optimization problems	[55]
Seagull Optimization Algorithm	Nature-inspired	Inspired by seagull migration and anti-predator behavior; uses exploration, exploitation, and avoidance strategies	Continuous optimization problems	[56]
Bayesian Optimization	Probabilistic	Uses Bayesian methods to model the objective function; employs Gaussian processes	Expensive black-box function optimization	[57]

## Data Availability

The data presented in this study are available on request from the corresponding author.

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
