# Peer review of "Optimized Machine Learning Models for Predicting Core Body Temperature in Dairy Cows: Enhancing Accuracy and Interpretability for Practical Livestock Management"

_animals, 2024, doi:10.3390/ani14182724_

Round 1
Reviewer 1 Report
Comments and Suggestions for Authors
This manuscript aimed at predicting the core body temperature (CBT) of dairy cows using different category features considering both environment-related and animal-related factors, and adopt SHAP-based feature analysis to improve the interpretability of black box models. This would be useful in contactless body temperature acquisition in dairy farm. There are some issues need the author's attention. According to the title, please clarify the enhancement of accuracy on predicting CBT and the major contribution in this study, comparing with the existing literatures.
Line 13:suggest to rewrite this sentence. machine learning is not a new approach to predict body temperature, and the models adopted in this manuscript is not that new.
Line 172-174: Infrared images were collected on the side of the cows' back in this study. Surface temperature on the cow back is greatly affected by the external environment, sprinkler, etc. Why don’t you choose a more stable area on the body surface (such as eyes, udder) or weighted calculation from multiple body surface area? Is there any existing research on the selection of monitoring area? Moreover, the description here is inconsistent with Line 227-232. Please clarify what is the part of body did the image collected in this study.
Section 2.1: Suggested to merge the subsection of ‘data acquisition’ and ‘feature extraction’, and then describe it in three parts: environment related, heat transfer related and animal related. This would be more convenient for readers to read and understand.
Line 304:please explain what is ANM. Abbreviation explanation is missing.
Line 302-310: The body temperature of cows is collected once at different time of the day, and the environment is collected once every 10 minutes. Please explain how are the data from different sources such as environment, body temperature and infrared images aligned in the dataset?
Section 3.1:As far as I see, there is no significance to do single factor comparison in the black box model. It is suggested to develop models in full factors considered and make comparison on the importance of all features to reduce input dimension.
Section 3.2:Any strategies adopted to prevent overfitting?
Line 511-519: In fig.13a, SHAP value significantly elevating from -0.4 to 0.55 when IRTave_TK range from 30 to 38. What does it mean when the SHAP value below zero? Please check, Fig.12(c) or Fig.13(c)?
Figure 11 and Line 573-576: the R2 is very low, what is the bottleneck for predicting CBT? Besides changing the inputs, did the authors tried any other way (eg. feature engineering, model integrating) to improve the prediction accuracy?
Figure 14: please explain what does it represents in the figure when the value is positive or negative.
Author Response
Comment 1: This manuscript aimed at predicting the core body temperature (CBT) of dairy cows using different category features considering both environment-related and animal-related factors, and adopt SHAP-based feature analysis to improve the interpretability of black box models. This would be useful in contactless body temperature acquisition in dairy farm. There are some issues need the author's attention. According to the title, please clarify the enhancement of accuracy on predicting CBT and the major contribution in this study, comparing with the existing literatures.
Response: As mentioned in the introduction, this paper enhances CBT prediction accuracy and has three main contributions: first, it evaluated different types of predictive variables and machine learning algorithms in detail; second, it explored the impact of intelligent optimization algorithms on model accuracy and parameter tuning efficiency; third, it used an advanced interpretation technique (SHAP) to facilitate understanding of model decisions. Please see L98-L107.
Comment 2: Line 13:suggest to rewrite this sentence. machine learning is not a new approach to predict body temperature, and the models adopted in this manuscript is not that new.
Response: Thank you for your suggestion. We have revised the wording. Please see L13-15.
Comment 3: Line 172-174: Infrared images were collected on the side of the cows' back in this study. Surface temperature on the cow back is greatly affected by the external environment, sprinkler, etc. Why don’t you choose a more stable area on the body surface (such as eyes, udder) or weighted calculation from multiple body surface area? Is there any existing research on the selection of monitoring area? Moreover, the description here is inconsistent with Line 227-232. Please clarify what is the part of body did the image collected in this study.
Response: As shown in Figure 3, we selected body regions such as the head, eyes, ears, cheeks, and udders as observation points for surface temperature. These selections were based on previous literature. Regarding your question about why we did not perform weighted calculations on the body temperature of these areas, to our knowledge, there is no standard formula for weighting surface temperatures in cows. The wording in our first draft was ambiguous, and it has now been revised to "side of the cows away from the fan" (L174).
Comment 4: Section 2.1: Suggested to merge the subsection of ‘data acquisition’ and ‘feature extraction’, and then describe it in three parts: environment related, heat transfer related and animal related. This would be more convenient for readers to read and understand.
Response: These two subsections correspond to the flowchart of our study. We separated feature extraction into a separate section to highlight the innovative aspects compared to previous research. If you insist on merging them, we will make the necessary revisions.
Comment 5: Line 304:please explain what is ANM. Abbreviation explanation is missing.
Response: Sorry for that, we have added the explanation for ANM. Please see L308-312.
Comment 6: Line 302-310: The body temperature of cows is collected once at different time of the day, and the environment is collected once every 10 minutes. Please explain how are the data from different sources such as environment, body temperature and infrared images aligned in the dataset?
Response: This is a rather detailed issue. In fact, we recorded the sampling time (accurate to the minute) and animal status when collecting the animal dataset (body temperature, infrared temperature). When constructing the dataset, we aligned the animal data with the environmental data through the sampling time.
Comment 7: Section 3.1:As far as I see, there is no significance to do single factor comparison in the black box model. It is suggested to develop models in full factors considered and make comparison on the importance of all features to reduce input dimension.
Response: Regarding your statement that "there is no significance to do single factor comparison in the black box model," we cannot fully agree. The single-factor comparisons are not performed within the black box model but rather after averaging the performance of six machine learning algorithms. The purpose of comparing different predictors is to achieve higher model prediction performance with fewer predictors. In practical production, obtaining all predictor variables is very challenging, and using all variables does not necessarily improve model performance. More importantly, there is evident high collinearity between these variables, not only within groups (e.g., IRTave_TK, ... IRTave_ER) but also across groups (e.g., Ta, THI, IRTave). Therefore, using all variables complicates model interpretation. This issue has been discussed in lines 593-596. As for your mention of "making comparisons on the importance of all features to reduce input dimension," this belongs to another interpretability technique—permutation feature importance.
Comment 8: Section 3.2:Any strategies adopted to prevent overfitting?
Response: This study mainly used two strategies to prevent overfitting: 5-fold cross-validation during model training (L321-322), and all algorithms included regularization hyperparameters in their tuning (Table 3).
Comment 9: Line 511-519: In fig.13a, SHAP value significantly elevating from -0.4 to 0.55 when IRTave_TK range from 30 to 38. What does it mean when the SHAP value below zero? Please check, Fig.12(c) or Fig.13(c)?
Response: SHAP values below 0 indicate negative contributions relative to the model's average output (BCT=38.856). This is reasonable because lower surface temperatures would reduce core temperature. Thank you for pointing out this error; it should be Fig.13 (c), and it has been corrected.
Comment 10: Figure 11 and Line 573-576: the R2 is very low, what is the bottleneck for predicting CBT? Besides changing the inputs, did the authors tried any other way (eg. feature engineering, model integrating) to improve the prediction accuracy?
Response: We believe the bottleneck lies in the complexity of the real production environment and the large individual differences among animals. Although the R2 value is "not ideal," it has improved compared to previous studies. The lower prediction error (MAE and RMSE<0.3℃) indicates that the model has practical value, as the accuracy of current electronic thermometers is about ±0.2℃. This study considered feature engineering techniques, and methods such as model ensemble and deep learning might slightly improve accuracy, which has been explained in the Limitations and Prospects section.
Comment 11: Figure 14: please explain what does it represents in the figure when the value is positive or negative.
Response: In the SHAP waterfall plot, positive values indicate that the feature's contribution increases the predicted value, while negative values indicate that the feature's contribution decreases the predicted value.
Reviewer 2 Report
Comments and Suggestions for Authors
The paper present interesting results. I emphasis the big dataset (n=3005), that allows to obtain reliable modelling results. IN general, the paper is structured and presented very well. All parts are presented clearly and concisely.
Here are my comments and suggestions:
Abstract: everything is ok, but probably should to report the full name of SHAP (SHapley Additive exPlanations).
Introduction: everything is ok
Materials and Methods: everything is ok, but here are some questions from my side.
Did you do some steps for reducing of multicollinearity of explanatory variables? Have you deleted highly correlated variables?
Results and discussion: everything is ok
Conclusion: ok
Author Response
Comment : The paper present interesting results. I emphasis the big dataset (n=3005), that allows to obtain reliable modelling results. In general, the paper is structured and presented very well. All parts are presented clearly and concisely.
Here are my comments and suggestions:
Abstract: everything is ok, but probably should to report the full name of SHAP (SHapley Additive exPlanations).
Introduction: everything is ok
Materials and Methods: everything is ok, but here are some questions from my side.
Did you do some steps for reducing of multicollinearity of explanatory variables? Have you deleted highly correlated variables?
Results and discussion: everything is ok
Conclusion: ok
Response: Thank you for your recognition of our work. We have added the full name of SHAP (L36). Regarding your concern about how to handle feature collinearity, we actually did not use all variables as input but individual variables to be evaluated combined with production-related variables and time zones as input, which are not highly correlated. This part has been discussed (L593-596).
Reviewer 3 Report
Comments and Suggestions for Authors
Overall, this is a very well-written manuscript which gives a really good example of data-driven modelling of core body temperatures in dairy cattle. I fully support this one to be published although I have some concerns for the authors to clarify in order to enhance its validity and implication for broader readers.
1. Your results are quite comparable with those reported by previous studies. It seems like further improvement is not easy. I guess it is impossible to achieve a predictive error even smaller than the camera's inherent error. If this is the case, further improvement of the data-driven model may be impractical. What directions, in your opinion, would future studies take in this context? This can be added to your discussion regarding future considerations.
2. L172. Where did you take infrared images? If in the barn, was the imaging affected by water sprinklers? If the surface is wet then the measured skin temperature will be biased downwards a lot. Could this have affected your results?
3. On L175, you mentioned that the accuracy of your camera is the best among all available choices. But accuracy is not the only key parameter, it is even less important than sensitivity and thermal resolution. Please check Guidelines for Veterinary Thermography. https://aathermology.org/wp-content/uploads/2018/04/Guidelines-for-Veterinary-Thermography-2022.pdf
4. Please explain all the parameters in your equations. These explanations in the text should be independent from tables. For instance, what is Rsw in equation (9)?
5. On L304. I miss here a definition of ANM possibly animal-based factors. Also the use of ANM is not clearly explained.
6. Figure 4. I am not sure if it is sensible to call this a white box since a white box usually comes with well-known theories.. this may be exaggerating. Please consider modifying.
7. L380, although you did not report the actual numbers of the results in this part, the simple feature set including ENV and ANM seems to have a good enough R2 and RMSE compared with those trained with much more complicated features and hyperparameter optimisation techniques. In this case, would this baseline model be more helpful for farmers? Those high-level features are not easy to access in the real world...
8. Figure 5, why there are no error bars like other figures?
9. Figure 6 on L397. So you added animal-related factors in these figures as well as the later ones. Please clearly explain this in M&M.
10. L408, the average temperature of the trunk was found the best. But in Figure 3 it is clear that this area was covered significantly by water. Could you explain why the average temperature is better than the maximum temperature in this case? The maximum temperature should, in theory, be less affected by water.
11. Have you tried training any models with all features combined? If not, why?
Author Response
Overall, this is a very well-written manuscript which gives a really good example of data-driven modelling of core body temperatures in dairy cattle. I fully support this one to be published although I have some concerns for the authors to clarify in order to enhance its validity and implication for broader readers.
Response: Thank you very much for the time and effort you have put into reviewing this article. Below are one-to-one responses to your specific comments.
Comment 1: Your results are quite comparable with those reported by previous studies. It seems like further improvement is not easy. I guess it is impossible to achieve a predictive error even smaller than the camera's inherent error. If this is the case, further improvement of the data-driven model may be impractical. What directions, in your opinion, would future studies take in this context? This can be added to your discussion regarding future considerations.
Response: In the previous draft, we have discussed possible future improvements in this area, which can be summarized as larger datasets, more powerful machine learning models, and more advanced parameter tuning algorithms. Specifically, model ensemble and stacking might slightly improve prediction accuracy. For continuous prediction of cow CBT, some deep learning models, such as long short-term memory networks, could be considered. Due to space limitations, the discussion of this part was not fully expanded. It is worth noting that, from the perspective of R2, the best model obtained in this study may not be entirely satisfactory, but the lower prediction error (MAE and RMSE<0.3℃) indicates that the model has practical value, as the accuracy of current electronic thermometers is about ±0.2℃.
Comment 2: L172. Where did you take infrared images? If in the barn, was the imaging affected by water sprinklers? If the surface is wet then the measured skin temperature will be biased downwards a lot. Could this have affected your results?
Response: The collection of infrared images of the cow's surface was conducted in the barns of commercial farms. The cows' surfaces were wetted by sprinklers, but the droplets evaporated quickly due to the fans before imaging took place. Spraying does affect the cows' CBT, but this does not impact our research, as our aim is to develop a CBT prediction model for individual cows in a real production environment.
Comment 3: On L175, you mentioned that the accuracy of your camera is the best among all available choices. But accuracy is not the only key parameter, it is even less important than sensitivity and thermal resolution. Please check Guidelines for Veterinary Thermography. https://aathermology.org/wp-content/uploads/2018/04/Guidelines-for-Veterinary-Thermography-2022.pdf
Response: Thank you for providing this information. We have checked the technical specifications of infrared imaging against the guidelines. We have revised the wording, please see L175-179, "This imager boasted a 336×252 image resolution, a thermal sensitivity of <50mk NETD (Noise Equivalent Temperature Difference) @30℃, a spatial resolution quality <1.27 mrad IFOV (Instantaneous Field of View), and an accuracy of ±2°C or ±2%, which meets the minimum specifications recommended in the Veterinary Thermal Imaging Guidelines [22]."
Comment 4: Please explain all the parameters in your equations. These explanations in the text should be independent from tables. For instance, what is Rsw in equation (9)?
Response: Added, please see the revised Table 2.
Comment 5: On L304. I miss here a definition of ANM possibly animal-based factors. Also the use of ANM is not clearly explained.
Response: Descriptions of the ENV feature set and ANM feature set have been added, please see L308-311, “ENV represents environmental variables such as air temperature, black globe temper-ature, relative humidity, and solar radiation, along with time zone data used as input features. ANS incorporates all the features from the first group plus additional varia-bles related to milk yield, days in lactation, parity, and body posture”。
Comment 6: Figure 4. I am not sure if it is sensible to call this a white box since a white box usually comes with well-known theories. this may be exaggerating. Please consider modifying.
Response: The image has been modified, removing "white box."
Comment 7: L380, although you did not report the actual numbers of the results in this part, the simple feature set including ENV and ANM seems to have a good enough R2 and RMSE compared with those trained with much more complicated features and hyperparameter optimisation techniques. In this case, would this baseline model be more helpful for farmers? Those high-level features are not easy to access in the real world...
Response: In fact, models trained with simple feature sets performed better than those trained with some derived feature sets, although they still lagged behind the best-performing feature set in terms of prediction accuracy. Note that Figure 5 shows the performance of six algorithms under ENV and ANM, while each x-axis in Figures 6-9 represents the average performance of the six models. We agree that the derived features are difficult to obtain, and part of our work involved exploring, trying, and comparing different features. This is also why we did not use all features for model training.
Comment 8: Figure 5, why there are no error bars like other figures?
Response: This is the result of a single algorithm under either the ENV or ANM feature set. Each column in Figures 6-9 represents the average result of the six algorithms.
Comment 9: Figure 6 on L397. So you added animal-related factors in these figures as well as the later ones. Please clearly explain this in M&M.
Response: We have added this content as per your suggestion, please see L318-320, “We assumed that animal-related variables were helpful for predicting individual BCTs in cows, so the IRTave group, IRTmax group, thermal comfort index group, and heat transfer group all included animal-related variables.”.
Comment 10: L408, the average temperature of the trunk was found the best. But in Figure 3 it is clear that this area was covered significantly by water. Could you explain why the average temperature is better than the maximum temperature in this case? The maximum temperature should, in theory, be less affected by water.
Response: In Figure 3(e), due to the color palette, it looks like this area is covered with water, but in fact, the temperature difference in this area was captured by infrared thermography. For your concern, please see L599-602, “IRTmax, as a temperature of a single pixel, has limited representational breadth, whereas IRTave provides a more comprehensive reflection of regional temperatures, responding more stably to external fluctuations”.
Comment 11: Have you tried training any models with all features combined? If not, why?
Response: We tried several algorithms, and using all features did not significantly improve prediction accuracy, which was not reported in the paper. As you mentioned earlier, some features are cumbersome to obtain. Most importantly, there is evident high collinearity between these variables, not only within groups (e.g., IRTave_TK... IRTave_ER) but also across groups (e.g., Ta, THI, IRTave). Therefore, using all variables complicates model interpretation. This issue has been discussed in L593-597.
Reviewer 4 Report
Comments and Suggestions for Authors
The paper is dedicated to noninvasive methods for estimating the core body temperature of dairy cows based on environmental and physiological data, infrared images, and machine learning algorithms. After a brief review of the scarce literature on the topic, the authors provide a detailed explanation of their approach, including a basic understanding of the ML tools and statistical methods involved. A special emphasis is placed on the data derived from the infrared images and their contribution to the model's accuracy. Time efficiency is also discussed and a comparison between algorithms is provided. Another strong feature of the article is the effort to improve interpretability, which is often lost in the complexity of neural networks and ensemble learning models. For that, the authors use SHAP analysis to assess the influence of individual features. This study could be useful in developing advanced noninvasive methods in farming and veterinary medicine.
Overall, the paper is well written, but needs a little improvement in some areas, e.g. the abstract is too detailed and technical which blurs the focus (no need to enumerate all the algorithms at that point). Sections 1 and 2 are fine, but a few more words could be said about the most closely related studies [14-17] to put the present work in perspective: Table 1 provides only partial information and later on in the manuscript [15] and [16] have been discussed in essence (page 24), but [14] and [17] have not. Apart from that, the text is quite well arranged, with rich graphic content and sufficient (even at points too much) detail. The English is also very good.
Author Response
Comment: The paper is dedicated to noninvasive methods for estimating the core body temperature of dairy cows based on environmental and physiological data, infrared images, and machine learning algorithms. After a brief review of the scarce literature on the topic, the authors provide a detailed explanation of their approach, including a basic understanding of the ML tools and statistical methods involved. A special emphasis is placed on the data derived from the infrared images and their contribution to the model's accuracy. Time efficiency is also discussed and a comparison between algorithms is provided. Another strong feature of the article is the effort to improve interpretability, which is often lost in the complexity of neural networks and ensemble learning models. For that, the authors use SHAP analysis to assess the influence of individual features. This study could be useful in developing advanced noninvasive methods in farming and veterinary medicine.
Overall, the paper is well written, but needs a little improvement in some areas, e.g. the abstract is too detailed and technical which blurs the focus (no need to enumerate all the algorithms at that point). Sections 1 and 2 are fine, but a few more words could be said about the most closely related studies [14-17] to put the present work in perspective: Table 1 provides only partial information and later on in the manuscript [15] and [16] have been discussed in essence (page 24), but [14] and [17] have not. Apart from that, the text is quite well arranged, with rich graphic content and sufficient (even at points too much) detail. The English is also very good.
Response: Thank you very much for taking the time to review this article and for recognizing our work. Regarding your comment about the abstract being too detailed, we have removed some descriptions of the optimization algorithms. We focused on comparing and analyzing our results with those in references [15] and [16], mainly because their datasets, research methods, and objectives are similar to ours. Reference [17] is relatively old, and the experimental content and details reported are not rich enough. The predictors and algorithms used in reference [14] differ from those in this study.
Round 2
Reviewer 1 Report
Comments and Suggestions for Authors
The authors responded and justified all points of the review. There are some minor mistakes ( eg. ANM or ANS?, not inconsistent ) in the paper. please make sure to proofread the manuscript again.
Author Response
Comment: The authors responded and justified all points of the review. There are some minor mistakes ( eg. ANM or ANS?, not inconsistent ) in the paper. please make sure to proofread the manuscript again.
Response: Thank you for your suggestion. The ANS (L310 & L393) has been corrected into ANM and highlighted.